# Update on Interventional Management of Neuropathic Pain: A Delphi Consensus of the Spanish Pain Society Neuropathic Pain Task Force

**DOI:** 10.3390/medicina58050627

**Published:** 2022-04-30

**Authors:** Ancor Serrano-Afonso, Rafael Gálvez, Elena Paramés, Ana Navarro, Dolores Ochoa, Concepción Pérez-Hernández

**Affiliations:** 1Pain Clinic, Department of Anesthesia and Reanimation, Hospital Universitari de Bellvitge, 08907 L’Hospitalet de Llobregat, Spain; 2Pain Clinic, Hospital Universitario Virgen de las Nieves, Avenida de las Fuerzas Armadas 2, 18014 Granada, Spain; rafaelgalvez@hotmail.com; 3Pain Clinic, Department of Anesthesia and Reanimation, Hospital Povisa, Rúa de Salamanca 5, 36211 Vigo, Spain; elenaparames@hotmail.com; 4Centro de Salud Puerta del Ángel, Paseo de Extremadura 113, 28011 Madrid, Spain; navarrosiguero@gmx.es; 5Clinical Pharmacology, Hospital de la Princesa, Calle Diego de León 62, 28006 Madrid, Spain; dochoa@iis-princesa.org; 6Head of Pain Clinic, Hospital de la Princesa, Calle Diego de León 62, 28006 Madrid, Spain; concha.phte@gmail.com

**Keywords:** neuralgia, interventional pain management, intractable pain, delphi technique, review

## Abstract

*Background and Objectives:* Interventional management of neuropathic pain (NP) is available to the patients who do not obtain satisfactory pain relief with pharmacotherapy. Evidence supporting this is sparse and fragmented. We attempted to summarize and critically appraise the existing data to identify strategies that yield the greatest benefit, guide clinicians, and identify areas that merit further investigation. *Material and Methods:* A two-round Delphi survey that involved pain clinic specialists with experience in the research and management of NP was done over an ad hoc 26-item questionnaire made by the authors. Consensus on each statement was defined as either at least 80% endorsement or rejection after the 2nd round. *Results:* Thirty-five and 29 panelists participated in the 1st and 2nd round, respectively. Consensus was reached in 20 out of 26 statements. There is sufficient basis to treat postherpetic neuralgias and complex regional pain syndromes with progressive levels of invasiveness and failed back surgery syndrome with neuromodulation. Radiculopathies and localized NP can be treated with peripheral blocks, neuromodulation, or pulsed radiofrequency. Non-ablative radiofrequency and non-paresthetic neuromodulation are efficacious and better tolerated than ablative and suprathreshold procedures. *Conclusions:* A graded approach, from least to most invasive interventions has the potential to improve outcomes in many patients with common refractory NP conditions. Preliminary promising data warrant further research on new indications, and technical advances might enhance the safety and efficacy of current and future therapies.

## 1. Introduction

Lesions or diseases of the somatosensory system can lead to increased pain sensitivity and spontaneous pain. A substantial portion of the population endures this neuropathic pain (NP) [1]. NP is a relatively prevalent condition. It varies between 6.9% and 10% depending on the tool used for its diagnosis [1]. And it is generally chronic, severe [2], very unpleasant and disabling [3,4,5]. Subsequently, it poses a considerable burden on the individual patients, healthcare resources and society [5,6]. Overall, theses burdens are owing to increased drug prescriptions and visits to healthcare providers [2] Unfortunately, NP disorders usually have a complex pathophysiology with limited correspondence between etiology, underlying mechanisms, and clinical manifestations [7,8]. This makes them difficult to manage [8], requiring specific treatment that involves a wide range of both pharmacological and interventional approaches [8,9,10].

Many patients on drug therapies receive modest or no relief [10]. Overall, there is a modest response to drug therapy. Many other experience unbearable side effects [10,11,12]. Under a more stringent, clinically sound testing scenario, the efficacy of current mainstream pharmacological approaches is lower than expected [12]. In addition, decreases in the effect of drugs for NP have been reported across all drug classes, with a progressive increase in Number Needed to Treat (NNT) values on randomized control trials (RCTs). Even among individuals with seemingly singular NP conditions, substantial diversity exists with respect to various clinical manifestations, sensory examination features and presumably underlying pain mechanisms [7]. Since NP perceived by patients is usually described with different negative and positive symptoms, and since different sensory profiles have been detected on the same patients, it is reasonable to think that pain relief may be difficult to achieve with single drug therapies. Clinical guidelines recommend starting treatment with monotherapy and place other interventions for those patients who do not respond to monotherapy or switching [10]. Some of these are not recommended as a clinically feasible treatment option, unless they are performed in hospital settings, under specialist care [9].

In addition, there has been little progress in NP pharmacotherapy during the last decade, which has not seen major innovations in the body of related scientific evidence [11]. In this context, the use of invasive interventions for NP is expanding [2,9,13]. However, the evidence supporting interventional treatments for NP is still sparse, heterogeneous, and fragmented; and they have received much less attention in evidence-based clinical practice guidelines than pharmacotherapies [14,15]. The last comprehensive international recommendations are almost 10 years old [9,15]. The data reviewed had considerable limitations and gaps which made it of limited applicability due to important shortcomings. Obtaining high quality evidence in this area is particularly challenging due to the very nature of the interventions. However, since the publication of the international guidelines, clinical experience has continued to expand, new technical developments have emerged, and several new studies providing low to moderate levels of evidence have been published. But in most cases the evidence provided was inconclusive [16]. Randomized controlled blind trials on interventional treatments are particularly hard to conduct because of practical, technical, and ethical difficulties [9]. This may partially explain why their efficacy and safety remain relatively unclear despite their therapeutic potential [17]. Reportedly, clinical practice guidelines have several shortcomings, particularly with respect to their applicability and translation into routine practice [14].

Thus, clinicians often find themselves making clinical decisions under these circumstances [16]. In the end, we find that clinical practice in different centers diverges because clinicians make decisions based on the techniques that each one believes has the strongest evidence. Given the lack of high-quality evidence on which to base recommendations, consensus statements can provide guidance to treating physicians to ground their practices and offer alternatives to patients enduring persisting pain [18]. The Spanish Pain Society (SED) embraced an initiative to support and contextualize current therapeutic practices for NP, and to help optimize current resources until new relevant developments emerge. The SED NP Task Force is entrusted with training its partners on NP and conducting research to improve the quality of the evidence on the various treatments for NP. As part of these efforts, the SED NP Task Force involved pain experts throughout Spain in a Delphi process to (a) clarify, organize, and align opinions on the general efficacy of interventional management of NP, (b) support the decision making on the specific procedures and indications that yield the greatest benefit, (c) provide a framework for reducing empiricism, and (d) identify promising strategies that merit further enquiry. The results are the focus of this paper. A complementary Delphi survey on the role of off-label pharmacological management NP was also carried out, but the results have been included in a separate article that is also been submitted for publication.

## 2. Materials and Methods

### 2.1. The Delphi Technique 

This technique is frequently used in medical research as a survey method to gain consensus [19]. It consists of a highly structured group interaction where members (called panelists) usually interact via questionnaires and receive feedback through facilitators. We followed some standards to strengthen the methodologic quality of this research, including [20]:the use of a reproducible procedure to select participants and a pre-specified definition of consensus (see below),a fixed criterion to stop the process after the completion of the second round regardless of the level of consensus reached,and the inability to drop, add or combine items between both rounds by neither study coordinators nor panelists.

To preserve anonymity and allow free expression of opinions, the panelists did not physically meet, but were able to see and comment upon other participants’ responses through the questionnaires of the second round (see Figure 1). Since this study focused on therapeutical decisions rather than nuanced definitions, a high level of agreement was fixed a priori to recognize consensus on accepting (80% or higher endorsement) or rejecting (20% or lower endorsement) the statements.

The questionnaire for the second round contained the same initial statements but was personalized by incorporating the individual panelist’s ratings in the first round together with the mean and median ratings of the entire panel for each item. All the statements had to be re-rated in the second round, either with the same rating as before or a modified rating in consideration of those from the other participants. The completion of the first round took about one month early in 2020. Then the study coordinators prepared the personalized questionnaires for the second round, which concluded six months after study initiation. This term was longer than expected because it was disrupted by the coronavirus pandemic (see the Results).

### 2.2. The Questionnaire

Respondents’ agreement was sought on whether particular procedures were appropriate for different NP conditions and the level of evidence supporting their use. For this purpose, the study questionnaire presented a series of statements to be rated on 5-point bipolar Likert scales of agreement (from 0 = strongly disagree to 4 = strongly agree). The study coordinators, who were members of the SED NP Task Force, used the available scientific literature (PubMed, Google Scholar, Web of Science and Scopus) as well as their personal research and clinical experience to develop it. This was done in two stages, generation, and consensus, in which candidate items were proposed and collated into a preliminary draft that was subsequently narrowed down in a series of consecutive rounds to reduce the number of items and agree on their final wording.

The final version (Table 1, Appendix A for original version in Spanish) included 26 items/statements divided into 4 sections (botulinum toxins, neuromodulation, radiofrequency, and infiltrations). Although some clinical guidelines consider botulinum toxins (BT) as a drug therapy, we addressed them in this survey because they must be administered by injections and in Spain it is only possible to use it after being prescribed under compassionate use in a hospital setting.

### 2.3. Participants

A database of pain clinic specialists providing care for patients in pain run by the SED was used to identify experts on NP research and management, which were approached by email with an invitation to participate in the Delphi survey study (see Appendix A for email examples). Candidates were not only searched for by publication indexes (i.e., h-index, number of papers or citations), but also by years of expertise in clinical practice, local or national key opinion leaders, etc.… The goal of this broad-spectrum selection was to be able to access knowledge from different perspectives to, in turn, be able to reach a more meaningful consensus within the community. Their identities were kept confidential throughout the study. The study coordinators prepared a set of scientific evidence that was circulated among participating panelists at least 4 weeks before the commencement of the Delphi process. This set comprised full texts of relevant articles and monographies, targeted reviews, extracts of published literature and results of pre-clinical and clinical studies related to the present research. The panelists used these and other available evidence besides their own clinical experience to answer the survey questions.

### 2.4. Ethical and Legal Aspects

The Clinical Research Ethics Committee of Bellvitge University Hospital in Barcelona approved the final study protocol, and all panelists provided written informed. The Spanish laws for the protection of personal data were observed during this research.

## 3. Results

A total, 97 identified candidates were contacted, of which only 35 agreed to participate and answered the first round. From them, only 29 participated in the second round. Information on participants is shown on Table 2. The consensus was moderate in the first round, as the panelists only agreed on 13 of 26 statements (50.0%). The sections on radiofrequency and infiltrations were the most controversial (see Figure 2, Appendix A). Consensus was higher in the second round, expanding to 20 out of 26 statements (76.9%). Nonetheless, the infiltrations remained contentious, as 3 out of 6 unsupported statements belonged to this section (see Figure 2, Appendix A). The consensus was always on accepting statements; none were rejected unanimously (Figure 2 and Appendix A).

### 3.1. Botulinum Toxins

There was a swift consensus that evidence supports BT efficacy and safety and recommends them for refractory NP, even in elderly patients, either alone or as an adjunctive therapy (Figure 2A, items 1, 2 and 4). In fact, the panelists acknowledged that some guidelines list them as third-line NP therapy despite it only being prescribed under compassionate use in Spain. They also agreed that the doses for typical intradermal/subcutaneous administration are highly variable, as they relate to the size of the target area and that, despite weak evidence, perineural blocks with these agents might be effective for some NP conditions (Figure 2A, items 6 and 7). Thus, further research into the later modality may be pursued. The size of the effect was more contentious; it was not until the second round that the panelists agreed it to be modest, accounting for large studies that showed conflicting results with previous, smaller positive trials (Figure 2A, item 3).

The consensus was not reached that BT are less effective in a subgroup of patients with a particular sensory phenotype that feature hypoesthesia or thermal sensory disorders (Figure 2A, item 5).

### 3.2. Infiltrations

There was much controversy in this area, as agreement was only achieved on 3 out of 6 (50.0%) statements. Consensus was quickly achieved that there is substantive evidence favoring the following points:that infiltrative blocks for postherpetic neuralgia (PHN) should be recommended in clinical practices (Figure 2B, items 8),that perineural blocks with corticosteroids are effective for NP in general (Figure 2B, 11),and that first lumbar spinal (ilioinguinal and iliohypogastric) nerve blocks with local anesthetics and corticosteroids in combination are effective to treat chronic postsurgical pain (CPSP) after hernia repair (Figure 2B, 13).

Although the trend was towards agreement, consensus was not reached on subcutaneous/intralesional injections of BT or corticosteroids and stellate ganglion blocks being recommended first for PHN, followed by paravertebral or epidural blocks. There was also no consensus on sympathetic blocks being used in general for this NP condition (Figure 2B, items 9 and 10). There was considerable disagreement, although consensus for rejection was not reached, that sympathetic blocks are ineffective for complex regional pain syndromes (CRPS) (Figure 2B, item 12).

### 3.3. Radiofrequency

Consensus was reached in 4 out of 6 statements (66.7%), and it was in the second round in all but one. The single item for which consensus was already attained in the first round stated that pulsed radiofrequency (PRF) is an effective alternative to treat PHN (Figure 2C, item 15). Already in the second round, panelists agreed that early application of PRF reduces herpes pain and the incidence of PHN, and that the latter condition would require repeated, long-lasting therapies (Figure 2C, items 16 and 17). Also in the second round, consensus was reached that conventional radiofrequency is more effective than PRF therapy for idiopathic trigeminal neuralgia (Figure 2C, item 18).

There was no agreement that radiofrequency is not indicated for CRPS, or that radiofrequency thermoablation of either the trigeminal ganglion or the trigeminal branches has similar effects (Figure 2C, items 19 and 20).

### 3.4. Neuromodulation

Consensus was reached in all items of this section (Figure 2D). Already from the first round, panelists agreed that stimulation of the dorsal columns of the spinal cord is indicated for treatment of failed back surgery syndrome (FBSS) and the CRPS (Figure 2D, item 20). They also unanimously saw the dorsal root ganglion (DRG) as an effective target for stimulation to treat radicular pain as well as localized NP (Figure 2D, items 22 and 23). Likewise, consensus that the stimulation of nerve fibers outside the neuroaxis (peripheral nerve stimulation) can be used for postsurgical or posttraumatic NP was swiftly reached (Figure 2D, item 24).

Agreement on the stimulation modes was somewhat more problematic; just 60.0% and 68.6% of panelists deemed in the first round that high frequency and burst patterns are as efficacious as conventional tonic stimulation for FBSS or NP in general, respectively (consensus was reached in the second round: Figure 2D, items 21 and 26). We also had to wait until the second round to obtain consensus that the minimally invasive percutaneous electrical nerve stimulation is helpful to treat chronic NP (Figure 2D, item 25). Other important conditions, such as painful diabetic neuropathy, PHN or spinal cord injury-associated pain were not covered in this survey, as was the case with intracranial neurostimulation.

## 4. Discussion

It is almost 10 years since the last comprehensive international guidelines for the interventional management of NP were released. Our objective was to provide an updated overview of all these issues by combining previous and new evidence with the critical evaluation and experience of experts within the formal structured framework provided by the Delphi process.

Some salient themes (see Table 3) were that (1) subcutaneous and even perineural BT should be formally considered in future guidelines as an add-on bridge treatment between pharmacotherapy and invasive interventions, (2) neuroaxial, sympathetic and peripheral blocks may prove effective (and should be attempted before resorting to more permanent interventions) for PHN, CRPS, postherniorraphy groin pain and other indications that should be further investigated, (3) (repeated) PRF can prevent and improve PHN and ameliorate CRPS, whilst the traditional ablative gasserian procedure is best suited for trigeminal neuralgia (TN), and (4) for the most recalcitrant cases, stimulation, even if nonparesthetic, of dorsal columns may be helpful for FBSS and CRPS, dorsal ganglia for radicular pain and more localized NP conditions, and peripheral nerves for posttraumatic and postsurgical pain and other potential peripheral indications. A more nuanced discussion on these issues follows.

The main, best-known mechanism of action of BT is the inhibition of the exocytosis of acetylcholine from cholinergic nerve endings [21]. However, they may also reduce pain by inhibiting the release of pain mediators from DRG and dorsal horn neurons (after retrograde transportation), the translocation of transient receptor potential vanilloid 1 to cell membranes and the trafficking of mechanosensitive channels in primary sensory neurons, as well as by decreasing local inflammation around nerve terminals and sympathetic transmission [21,22]. Indeed, they have been successfully tested in frequent peripheral NP conditions [23,24,25] and even in spinal cord injury-related pain [26], and, as panelists acknowledged, have been endorsed as third-line therapies by some guidelines [10] despite the available studies generally being small and of low to moderate methodological quality [10,23,24,25,26]. In consistence with published data [22], panelists also agreed that analgesic doses are highly variable. There was controversy regarding the size of the effect of BT on NP. The evidence is contradictory in this regard, since the considerable effects reported in small studies [24] have not been corroborated by larger ones [10,27]. Small effect sizes would justify them being recommended as add-on therapies, as panelists conceded, despite little attention being given to this in the literature [28]. Consensus was not reached on the existence of certain less responsive phenotypes including hypoesthesia and thermal deficits. Only one high-quality study has suggested this [27]. In addition to the traditional subcutaneous route, occasional assessments of the effects of perineural administration of BT have yielded encouraging outcomes [29,30], which were acknowledged by the panelists.

The interruption (block) of neural conduction by the injection of anesthetic, anti-inflammatory or neurolytic agents to treat pain has a long history [31], yet it is not devoid of controversy. Although neurolytic procedures may have serious complications, such as pain worsening or bringing new pain syndromes, transient blocks would avoid complications related to deafferentation but would have short-lived effects and require repeated doses for prolonged pain relief [32]. Furthermore, although the effects of neural blocks are of considerable magnitude and consistency, related evidence is scarce and of low quality [33,34]. Both somatosensory and sympathetic blockades with local anesthetics and/or corticosteroids in the acute or subacute phases of herpes zoster episodes have proven to be effective in reducing the incidence of PHN [9,35]. The attenuation of central sensitization secondary to repetitive pain signaling, as well as neural inflammation and sympathetically mediated vasospasm may explain this incidence reduction when injections are administered in the paravertebral [36] or epidural spaces [35].

Less clear are the effects when these blocks are administered once PHN has developed. Although traditionally discouraged [9], paravertebral blocks are still considered a therapeutic option for this condition despite the lack of evidence in their favor [34]. This uncertain background could explain the varied responses of the panelists, who supported their clinical use and did not discard sympathetic blockade for PHN but did not agree with a proposed sequence of interventions that placed subcutaneous injections of BT ahead of paravertebral or epidural blocks. The single intervention for which there is high-quality (but conflicting) evidence, the intrathecal methylprednisolone injection [34], is a fourth line treatment that was not addressed in this study. The panelists widely agreed with the statement that perineural corticosteroids reduce NP. This statement is quite generic and does not differentiate between neuraxial and peripheral administration or between injections and continuous infusions. There is ample evidence supporting the use of epidural corticosteroid injections for neuropathic spinal pain [37] and, to a lesser extent, the use of perineural corticosteroid blocks for compression, traumatic or iatrogenic peripheral nerve injuries [38,39], and some specific prevalent conditions such as PHN [9,40] or carpal tunnel syndrome [41]. Nevertheless, the quality of the evidence is generally low. Other types of blocks, such as plexus blocks featuring steroids, have received attention from a perioperative perspective or in the context of chronic non-NP. The certainty of the response of the panelists could stimulate further methodologically sound studies in these conditions or areas that have not been evaluated from a neuropathic perspective. Likewise, panelists clearly endorsed the effectiveness of ilioinguinal and iliohypogastric blocks for the treatment of chronic postherniorrhaphy groin pain. There is evidence that such blocks can reduce acute postoperative pain [42]. However, although these nerves are thought to be involved in the pathogenesis of CPSP [43], it is unclear whether blocking them is effective in the treatment of this form of NP [44], but panelists’ responses favored it. On the contrary, the statement that sympathetic blocks are ineffective for CRPS was almost unanimously rejected. Consistently, sympathetic blocks have been considered the first line interventional treatment option for such syndromes and may be used to assess the appropriateness of future, more permanent neurolytic or neuromodulatory procedures [9,45].

Radiofrequency is a percutaneous procedure that is used to ablate or modulate neural structures using generally less invasive techniques than electrical neuromodulation. A localized electromagnetic field is delivered from a catheter needle tip that produces varying degrees of thermo-denaturation lesioning and electrical stimulation of (neural) tissues as the temperature increases and electrical fields that are produced around the tip of the needle [46,47]. The thermal effects are due to molecular oscillation caused by the alternating electrical field, whilst the electric effects are thought to be mediated by disruption of ion channels and transmembrane potentials that result in conditioning of sensitive afferents, reducing their synaptic efficiency at their central relays [46,48]. Thermal lesioning produces the destruction of neural tissue such that an effective cordotomy, rhizotomy or neurotomy takes place. It was originally used to treat trigeminal neuralgia and spinal pain (e.g.,) [49]. Following the initial thermoablative application, the discovery of the therapeutic potential of the electric effects themselves led to the development of PRF, in which short bursts of radiofrequency energy are delivered separated by pauses to allow heat to dissipate and thus avoid permanent lesioning of tissues [47].

The panelists quickly endorsed that PRF is effective for PHN in general. This agrees with some studies that have shown its efficacy when applied over the DRG or intercostal nerves from the cervical to lumbosacral areas [34,50]. The evidence about the effects on postherpetic TN is, nevertheless, much smaller [51]. It was also agreed that early application of PRF during the acute or subacute phases of herpes zoster could reduce pain and incident neuralgia, as some authors have proposed [51,52,53]. In contrast, the panelists acknowledged that PRF may be less effective for idiopathic TN, for which conventional thermal radiofrequency lesioning is preferred [50,54,55]. However, the latter can cause bothersome complications associated to neurolysis [55,56]. Selective lesioning of the trigeminal peripheral branches has been proposed to avoid such complications [57,58]. Panelists did not agree on this strategy, even though the evidence in this respect is only against pulsed, not thermo-ablative radiofrequency [59]. Combining pulsed and conventional radiofrequency to reduce the thermal dose delivered has also been proposed to reduce complications [55]. Commonly, studies either had short follow-up periods or showed considerable recurrence rates after 1 year [34,55,60]. Consistently, the panelists reached consensus that repeated radiofrequency treatments should be considered for PHN.

On the other hand, the use of radiofrequency for CRPS was controversial. Nearly half of the panelists rejected the idea that it should not be used for these conditions. The conceptualization of these syndromes as sympathetically maintained pains led to the consideration of sympatholysis among the available therapeutic options [61]. However, there is only sporadic evidence favoring the use of radiofrequency sympathectomy for CRPS and concerns about relevant complications, like those described for neurolytic blocks, limit their use [62,63,64]. PRF has been successfully used to avoid ablation-related complications, but its effects appear to be short-lived [64].

Neuromodulation is the alteration of nerve activity by delivering electrical or pharmaceutical agents directly to targeted sites of the body [65]. Due to its invasiveness, it is currently recommended as a fourth-line NP treatment after failure of pharmacotherapy and other interventional therapies [13]. It involves two types of treatments, electrical and chemical [65]. The latter generally refers to intraspinal or intraventricular drug delivery through implantable pumps, which are modalities of intrathecal drug delivery that were not covered in this study. Electrical treatments, which have been the subject of renewed interest in recent years [66], typically involve the delivery of highly modulated and structured electrical currents through electrodes placed around precise structures of the nervous system, commonly the dorsal columns of the spinal cord, the DRG, or nerves outside the neuroaxis [18]. One of the first clinical electrical neuromodulation paradigms was based on replacing pain with more bearable paresthesia through the activation of low threshold large sensory fibers within the dorsal columns by low frequency tonic suprathreshold currents that are known to reduce transmission of pain signals in the dorsal horns via both segmental and supraspinal pathways [67]. The panelists’ views were consistent with mounting evidence and agreed that this type of spinal cord stimulation (SCS) is effective for CRPS and FBSS [18,68]. The success of newer stimulation strategies involving the use of high frequency currents or burst delivery patterns to reduce pain without causing paresthesia has widened the spectrum of putative mechanisms of action and the indications of spinal cord stimulation [18,67,69].

However, in consonance with the more rudimentary understanding of these newer modalities, the related items were more controversial as panelists did not gain consensus until the second round. The relatively defined anatomical projections and the active role that DRG have in pain processing and transmission has made them an attractive target for electrical neuromodulation [18]. In recent years, some studies have shown that DRG stimulation can be as effective or even more effective than SCS in several peripheral NP conditions, particularly when they are unilateral, and areas of pain do not span more than 2 dermatomes [70]. Unlike new SCS strategies and despite the evidence also being sparse, the consensus on DRG stimulation to treat peripheral localized NP was already attained in the first round [69]. Consensus was also immediate that this treatment should be considered for radicular pain, although the evidence on this topic is almost nonexistent. Given the favorable opinions of the panelists on DRG stimulation and the absence of high-quality evidence in NP conditions other than CRPS, further high-level research in this area is encouraged, as others have also pointed out [71]. Long-term complications associated with lead migration and fracture should be addressed as well [72]. Direct electrical stimulation of peripheral nerves was proposed to treat pain even before SCS, but its use stalled in the past mainly due to its invasiveness [18]. Recent technological advances have allowed less invasive methods of electrode placement, and several studies have evaluated its application in several NP conditions [73]. Positive high evidence grade studies have been reported in patients with peripheral NP secondary to trauma or surgery, postamputation pain and lower back pain [73]. The panelists endorsed the use of peripheral nerve stimulation for posttraumatic or postsurgical neuralgia already in the first round but required both rounds to achieve consensus in other applications.

There are some weaknesses and biases to this work that must be commented. However, we tried to minimize its impact. First, the production of this consensus in middle 2020 could have affected the results. We have no evidence about this, but it must be considered. The production of the Delphi consensus was started before pandemic outbreak (see Appendix A). The pandemic outbreak probably affected the number of the identified experts who finally wanted to participate in the survey, as the first round was launched immediately before the outbreak arrived in Spain. A broad-spectrum selection was performed with the intention to achieve consensus from experts with different perspectives. Initially 97 experts from different fields and profiles were identified. But only 35 agreed to participate. This, most probably had to do with the pandemic outbreak which was concurrent in time. Nevertheless, we decided to follow on with the process, even though this could distort results due to lack of participation. More panelists could have changed or strengthened the results. For this, a second Delphi process will be considered to be carried out in the coming years. Fortunately, for the ones who did agree to answer, we only got a delay in responses for the 1st round, and some panelist dropouts from 1st to 2nd round. The drop in participation from first to second round is attributed to panel attrition between the two rounds, again due to second wave of the outbreak in Spain.

Secondly, although the construction of the questionnaire was inclusive, dissenting opinions tend to be ruled out during the item reduction phase for the sake of reaching a consensus [74]. To avoid it, the panelists had to focus on the set of statements contained in the survey, which necessarily limits the spectrum of therapeutic alternatives evaluated. This structural bias may also affect the Delphi process itself, as feedback between rounds may prompt agreement with choices that have a favored majority. Acquiescence biases can also affect respondents’ opinions because endorsing statements is easier than rejecting them [75]. We tried to minimize them by fixing the objectives, consensus criteria and number of rounds ahead of time, and by barring item modification, deletion, addition or combination by either study coordinators or panelists between both rounds [20].

Finally, we did not contemplate adding clinometric properties (i.e., measures of reliability and validity of the questionnaire) in our Delphi Study because we did an ad hoc questionnaire, limited to the context of the study. It makes little sense to go to the effort of measuring inter-observer reliability for a questionnaire that is not going to be used outside the study group). Even so, the questionnaire could have been validated externally by independent experts to strengthen its external validity. Jünger et al. [76] recommend considering the possibility of doing an external validation of the questionnaire by independent experts before proceeding with the Delphi. However, this was difficult because of limited resources and, after starting the Delphi process, the pandemic outbreak situation made contacting with other peers even harder.

Controversies have surrounded interventional pain medicine because of unclear long-term benefits and the relative disregard of psychosocial pain components [77]. Nonetheless, undeniable benefits can derive when it is used within the context of a comprehensive multidisciplinary approach [13,78]. Limitations in the available evidence and therapeutic effects [9] make it hard to derive convincing conclusions even after experts have arrived at a wide consensus. Although we have identified and commented on some interventions that are particularly suitable for specific indications, we cannot provide systematic guidance on which, how, and when they should be used for each condition. Even though, not all the statements gathered the same force of agreement. Some of them did not only achieve a higher level of consensus, but the variability coefficients were reduced between one round and the other (see Appendix A). The statements with the highest consensus can be seen in the following table (see Table 3).

## 5. Conclusions

This Delphi consensus and review suggest that a stepped approach to interventional management of NP based on invasiveness is feasible. Blocks can precede more permanent procedures, and non-paresthetic, non-neurolytic methods could attain good results while avoiding some undesired effects of ablation. There is a fair basis to treat some conditions such as PHN, CRPS, and FBSS, which account for much of the NP refractoriness, but high-grade studies are still required to corroborate them. Scientific background and promising results also warrant future research on peripheral blocks and neurostimulation, or PRF of the DRG for localized conditions and radiculopathies. The development of new adaptive and physiologically coherent electrical stimulation patterns may also be pursued.

## Figures and Tables

**Figure 1 medicina-58-00627-f001:**
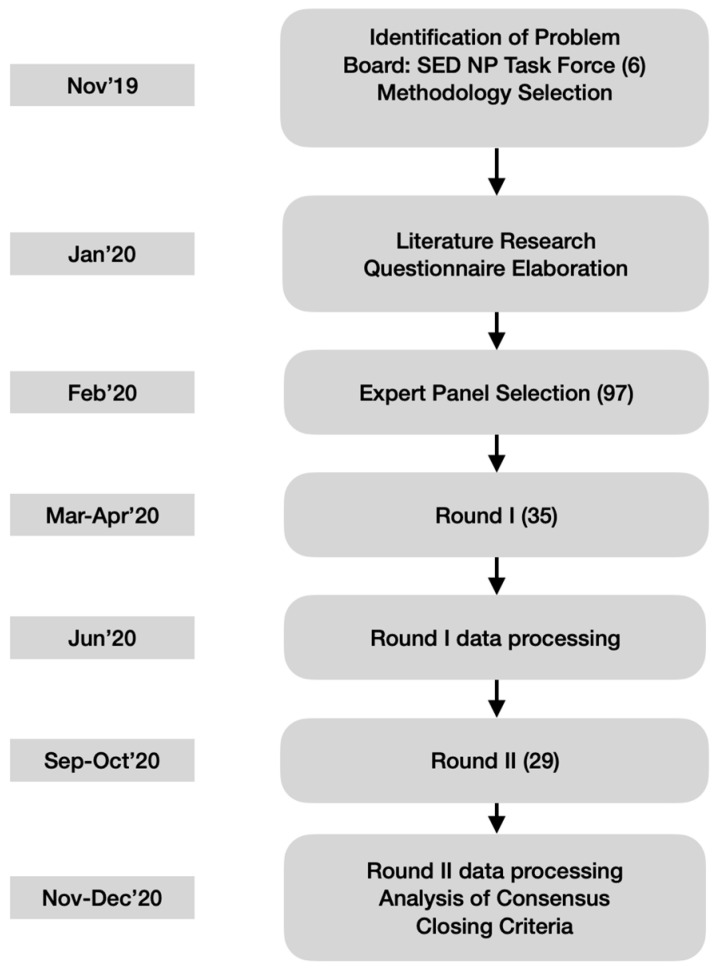
Flow Chart of the Delphi Technique applied. Timeline on left boxes. In parenthesis the number of people involved. 97 experts had been identified but only 35 agreed to participate in the Delphi Survey initially. Only 29 finished the second round. SED NP Task Force = Spanish Pain Society Neuropathic Pain Task Force. Total of 6. Authors.

**Figure 2 medicina-58-00627-f002:**
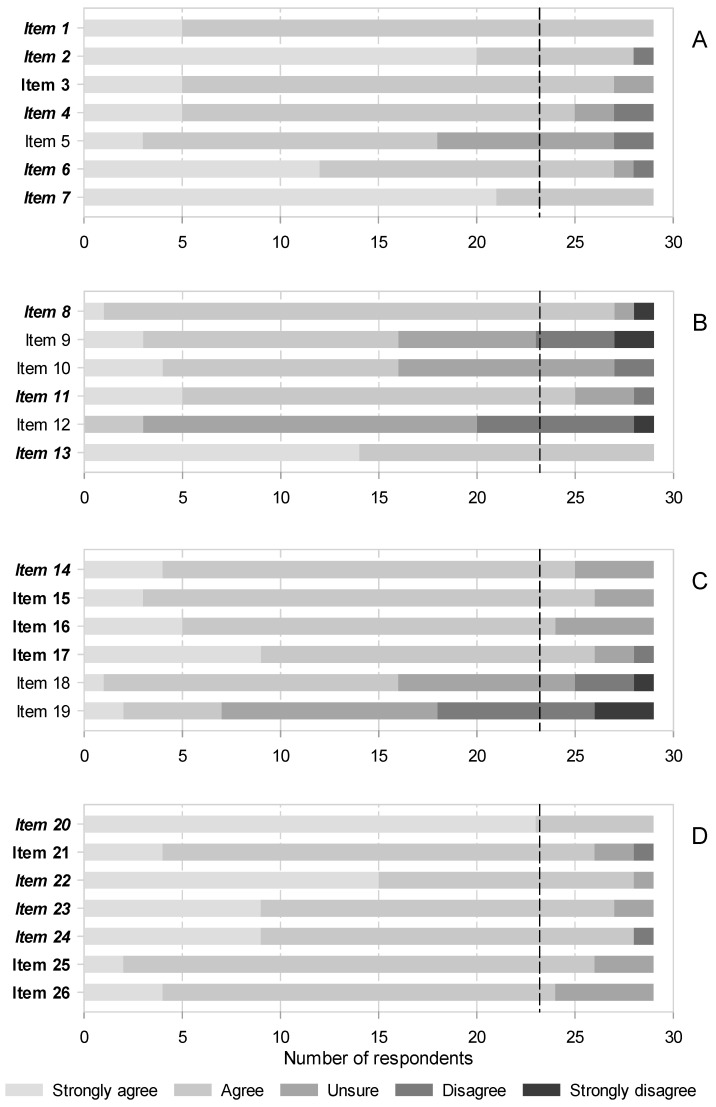
Agreement after 2 rounds on items related to interventional therapies for NP. (**A**) Botulinum toxins, (**B**) Neural blocks and infiltrations, (**C**) Radiofrequency, (**D**) Neuromodulation. See Table 1 for a description of the items. Italics and bold letters indicate the items (statements) for which consensus was reached in the first and second rounds, respectively. The vertical black dashed line indicates the predefined threshold to recognize consensus on accepting the statements (>80% of respondents). There was no consensus on rejecting any item.

**Table 1 medicina-58-00627-t001:** Description of the Delphi survey statements.

**(A) Botulinum toxins**
There is evidence about the efficacy and safety of botulinum toxins (BT) and, despite being under compassionate use, they are considered a third-line treatment by many NP guidelines. Its use in combination with standard therapy is recommended in refractory patients. The NNT is unknown and, given the small size of the studies, could be very low (lower than 2), although some larger, methodologically sound studies report values higher than 7. The BT is useful and efficacious in elderly patients with refractory NP. BT seems to be less useful in patients with hypoesthesia and thermal sensory disorders, so theoretically they are not good candidates for BT therapeutic assays. Therapeutic doses can range from just a few to as many as 200 units and are primarily related to the size of the painful area. The evidence on the use of BT in perineural areas is very weak (case series), although we cannot rule out its usefulness based on current publications. Therefore, embarking in rigorous studies to demonstrate the safety and efficacy of this promising new therapeutic modality is warranted.
**(B) Neural blocks and infiltrations**
8. Nerve blocks, or infiltrations, have level 2 evidence, grade B recommendation for the treatment of PHN. 9. The subcutaneous injections of BT/corticosteroids or the blockade of the stellate ganglion (facial herpes without PHN) are first level therapies, whilst paravertebral or epidural blocks and nerve blocks with steroids (for symptomatic relief) are second level therapies for PHN. 10. The recommendation is against the use of sympathetic blocks for PHN lesions. 11. Perineural corticosteroids reduce NP. 12. Sympathetic blocks with local anesthetics are not effective for CRPS. 13. Ilioinguinal and ilio-hypogastric blocks with local anesthetics and corticosteroids in combination can be effective for the treatment of chronic post herniorrhaphy groin pain.
**(C) Radiofrequency**
14. PRF can be considered an efficacious alternative for the treatment of PHN. 15. Repeated, long duration therapies should be considered for PHN. 16. Early application of PRF reduces both herpes pain and the incidence of PHN. 17. Conventional radiofrequency is more effective than PRF for idiopathic trigeminal neuralgia. 18. Thermal radiofrequency has similar results on the trigeminal ganglion than its peripheral branches. 19. Radiofrequency cannot be recommended for CRPS.
**(D) Neuromodulation**
20. Dorsal column stimulation therapy (SCS) is indicated for FBSS and CRPS. 21. High frequency (HF10) is as efficacious as low frequency stimulation for FBSS. 22. DRG stimulation should be considered for radicular pain. 23. DRG stimulation would be indicated for localized NP. 24. Peripheral nerve stimulation may be used for postsurgical or posttraumatic peripheral NP. 25. Percutaneous electrical nerve stimulation is useful for chronic NP. 26. BURST stimulation can be as efficacious as tonic stimulation for the treatment of NP.

BT: botulinum toxin, CRPS: complex regional pain syndrome, DRG: dorsal root ganglion, FBSS: failed back surgery syndrome, HF: high frequency, NNT: number needed to treat, NP: neuropathic pain, PHN: postherpetic neuralgia, PRF: pulsed radiofrequency, SCS: spinal cord stimulation.

**Table 2 medicina-58-00627-t002:** Information on Participants.

Participants	First Round	Second Round
Gender (M/F)	14/21	10/19
Years of experience in NP–mean (s.d.)	16.37 (8.81)	15.79 (9.16)
Dept Head/Director	9	7
Specialty		
Anesthesiology	23	20
Ph. Med. and Rehabilitation	9	6
Rheumatology	3	3
GP	1	1

Information on participants on de Delphi Survey, with years of experience on neuropathic pain, field of knowledge (i.e., specialty before going into pain practice), gender, and number of participants who hold a position of Department Head or Director. M = male. F = female. NP = neuropathic pain. s.d. = standard deviation. Ph. Med. = physical medicine. GP = general practitioner.

**Table 3 medicina-58-00627-t003:** Statements with highest level of consensus.

Statement	Mean Consensus Degree	Variability Change
Its use in combination with standard therapy is recommended in refractory patients. *	3.42/3.62	0.19
The evidence on the use of BT in perineural areas is very weak (case series), although we cannot rule out its usefulness based on current publications. Therefore, embarking in rigorous studies to demonstrate the safety and efficacy of this promising new therapeutic modality is warranted.	3.51/3.72	0.12
Ilioinguinal and ilio-hypogastric blocks with local anesthetics and corticosteroids in combination can be effective for the treatment of chronic post herniorrhaphy groin pain	3.46/3.48	0.15
Repeated, long duration therapies should be considered for PHN. $	2.74/3.00	0.15
Conventional radiofrequency is more effective than PRF for idiopathic trigeminal neuralgia.	3.09/3.17	0.22
Dorsal column stimulation therapy (SCS) is indicated for FBSS and CRPS	3.60/3.79	0.11
High frequency (HF10) is as efficacious as low frequency stimulation for FBSS	2.71/3.00	0.2
DRG stimulation should be considered for radicular pain	3.37/3.48	0.16
DRG stimulation would be indicated for localized NP	2.86/3.24	0.18
Percutaneous electrical nerve stimulation is useful for chronic NP	2.86/2.97	0.14

Statements that achieved the highest improvement of consensus with a reduction in the variability coefficient (Cv). Numbers in the column of “ Mean consensus Degree” reflect the mean between “0—Completely disagrees to “4”—Completely agrees. Horizontal lines differentiate between sections. Light-grey background is placed to differentiate between statements. Statements are written as a translation of the original, * speaking about Botulinum toxin (BT). $ speaking about radiofrequency. PHN = Post-herpetic Neuralgia. PRF = Pulsed Radiofrequency. FBSS = failed back surgery syndrome. CRPS = Complex Regional Pain Syndrome. DRG = dorsal root ganglion. NP = Neuropathic Pain.

## Data Availability

Data supporting reported results can be found, by request, to the Spanish Pain Society (SED) and/or to the SED Neuropathic Pain Task Force.

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
