# Peer review of "Update on Interventional Management of Neuropathic Pain: A Delphi Consensus of the Spanish Pain Society Neuropathic Pain Task Force"

_medicina, 2022, doi:10.3390/medicina58050627_

Round 1
Reviewer 1 Report
1) There are some grammatical English errors throughout the manuscript. Not commonly used, repetitive, and long structures. E.g.: ‘‘who do not attain satisfactory outcomes’’ ‘‘strategies that yield maximum benefit, orient clinicians, and identify areas that merit further investigation.’’ ‘‘To help optimize current resources until new relevant developments emerge, the Spanish Pain Society (SED) embraced an initiative to support and contextualize current therapeutic practices for NP, in particular the use of off-label drug therapies and interventional procedures.’’
2) Delphi consensus usually represents recent scientific evidence based on specialist opinions. Do the authors believe that the production of this consensus in middle 2020 can affect the results? Could the authors provide evidence for similar results?
3) It is advised a flowchart. Cascella, M.; Miceli, L.; Cutugno, F.; Di Lorenzo, G.; Morabito, A.; Oriente, A.; Massazza, G.; Magni, A.; Marinangeli, F.; Cuomo, A.; on behalf of the DELPHI Panel. A Delphi Consensus Approach for the Management of Chronic Pain during and after the COVID-19 Era. Int. J. Environ. Res. Public Health 2021, 18, 13372. https://doi.org/10.3390/ijerph182413372
4) Questionnaire.
a) It is advised to give a reference for each question or statistical analysis for its use. The clinimetric of questions can be provided as supplementary material.
b) Was the questionnaire in Spanish or English? It is advised to provide supplementary material in the original language.
5) Statistical analysis. It is missing this section.
6) Participants.
a) It would be interesting to provide something more information related to the specialist like their h-index, number of papers, or citations. This can be done as a table of baseline characteristics.
b) An example of the e-mail sent to participants it is advised provided. This can be a screenshot. Supplementary material.
7) Discussion
a) A table with the main results should be done. Highlights should be described. What is in agreement with the literature? What did the questionnaire achieve more than 80% of agreement? What did the questionnaire achieve agreement in less than 20%?
Author Response
Dear reviewer 1. Thank you very much for your time and dedication. I have attached a word document with replies to your comments. The new manuscript is in a word version that has all changes marked, as requested. However, to facilitate reading, we have included a pdf version with all the changes already accepted.

Reviewer 2 Report
Dear Authors,
In this manuscript, the authors attempted to summarize and critically appraise the existing data on interventional management of neuropathic pain (NP) to identify strategies that yield maximum benefit, orient clinicians, and identify areas that merit further investigation. A two-round Delphi survey that involved pain clinic specialists with experience in the research and management of NP was done over an ad hoc-item questionnaire prepared by the authors. Based on their finding, warrant future research on peripheral blocks and neurostimulation or PRF of the DRG for localized conditions and radiculopathies. The development of new adaptive and physiologically coherent electrical stimulation patterns may also be pursued.
The topic is timely and may attract much attention. I found the report quite innovative, very interesting, and scientifically sound. 
I have only one suggestion to improve this paper:
The introduction could be a bit more detailed, more could be written about NP and current therapeutic options.
Recommendation of revision: minor
Author Response
Dear reviewer 2. Thank you very much for your time and dedication. I have attached a word document with replies to your comments. The new manuscript is in a word version that has all changes marked, as requested. However, to facilitate reading, we have included a pdf version with all the changes already accepted.

Round 2
Reviewer 1 Report
There are some grammatical errors throughout the manuscript. e.g.
L.49 ''theses burdens is owing to increased drug''